# The Potential Antipyretic Mechanism of Ellagic Acid with Brain Metabolomics Using Rats with Yeast-Induced Fever

**DOI:** 10.3390/molecules27082465

**Published:** 2022-04-11

**Authors:** Fengfeng Xie, Liba Xu, Hua Zhu, Yaling Chen, Yinlan Li, Lizhen Nong, Yanfang Zeng, Sijie Cen

**Affiliations:** 1School of Chemistry and Chemical Engineering, Guangxi University for Nationalities, Nanning 530006, China; 15177143553@163.com; 2Collaborative Innovation Center of Zhuang and Yao Ethnic Medicine, Guangxi Key Laboratory of Zhuang and Yao Ethnic Medicine, Guangxi University of Chinese Medicine, Nanning 530200, China; xuliba15078772841@163.com (L.X.); xingyun170210@163.com (Y.C.); yinlan.li@beigene.com (Y.L.); nonglizhen10086@163.com (L.N.); yanfang.zeng@sanofi.com (Y.Z.); censijie0211@163.com (S.C.)

**Keywords:** fever, metabolomics, UPLC-MS, ellagic acid, biomarkers

## Abstract

Fever is caused by an increase in the heat production process when the body is under the action of a heat source or the dysfunction of the temperature center. Ellagic acid (EA) is a polyphenol dilactone that has anti-inflammatory, anti-tumor, and antioxidant activities. Male Sprague-Dawley rats were injected yeast to reproduce an experimental fever model (150 ± 20 g), and the rectal temperature and its change values were subsequently taken 19 h later; the excessive production of interleukin-1β (IL-1β) and prostaglandin2 (PGE2) induced by yeast was regulated to normal by EA administration. Rat brain metabolomics investigation of pyrexia and the antipyretic anti-inflammatory effect of EA was performed using Ultra-High-Performance Liquid Chromatography–Mass spectrometry (UPLC-MS). Twenty-six metabolites, as potential biomarkers, significantly altered metabolites that were found in pyretic rats, and eleven metabolites, as biomarkers of the antipyretic mechanism of EA, were significantly adjusted by EA to help relieve pyrexia, which was involved in glycerophospholipid metabolism and sphingolipid metabolism, etc. In conclusion, potential metabolic biomarkers in the brain shed light on the mechanism of EA’s antipyretic effects, mainly involving metabolic pathways, which may contribute to a further understanding of the therapeutic mechanisms of fever and therapeutic mechanism of EA.

## 1. Introduction

Fever is the most commonly observed symptom caused by infection or inflammation in many diseases [1], its response is coordinated by the central nervous system through endocrine, nervous, immune and behavioral mechanisms [2,3]. Among animal models of fever, yeast-induced pyrexia in rats is the most widely used. The occurrence of pyrexia involves numerous nerval routes and factors, such as IL-1β and PGE2. The level of IL-1β and PGE2 increases significantly in the animal fever model; this finding implies that the levels of these factors could be used to evaluate the antipyretic effect of drugs [4,5,6].

Aspirin, acetaminophen, and Ibuprofen (IB) are widely used drugs in the treatment of inflammation, fever, and pain. The mechanism of action results in the inhibition of key clinical signs such as redness, warmth, pain, and swelling [7]. However, accumulating evidence has demonstrated that these drugs can induce significant side effects [8].

Ellagic acid is a polyphenol dilactone; it not only exists in the form of dissociation, but also exist in many plants and fruits in nature in the form of condensation, and has been reported to have anti-inflammatory, anti-tumor, and antioxidant activities. EA shows its anti-inflammatory effects via its modulatory effect on cyclooxygenase (COX) enzyme, IL-6, TNF-α, and IL-1β production [9]. EA exhibits a potent anti-inflammatory effect against carrageenan-induced inflammation by the reduction in IL-1β, NO, MDA, TNF-α, COX-2, and NF-κB expression and induction of GSH and IL-10 production [10]. It has been shown that EA inhibits COX-2 and iNOS protein expression, and reduces the levels of IL-1β and PGE2 in lung tissues of OVA-sensitized/challenged mice, suggesting significant effects on airway inflammation [11].

Metabolomics analysis is used to detect and screen out metabolites with important biological significance and statistically significant difference from biological samples, and on this basis to clarify the metabolic process and change mechanism of organisms [12]. It can be used to analyze metabolic pathways or metabolic networks, and study the metabolic basis of macroscopic phenotypic phenomena of different organisms and the response mechanisms of metabolites stimulated by physical, chemical, or pathogenic organisms, such as diseases and drugs, and the safety evaluation of food and drugs [13,14]. Chromatography–mass spectrometry realizes the whole process from chromatography-separation to mass spectrometry identification. Liquid chromatography–tandem mass spectrometry (LC-MS/MS) can be used for accurate qualitative and quantitative analysis [15,16]. Metabolic research on fever has usually focused on bio-fluid organization samples, such as serum, plasma, urine, and brain [17,18,19,20]. However, fever is caused by the increase in the heat production process when the body is under the action of a heat source or the dysfunction of the temperature center; the brain’s thermoregulatory center plays a key role. The extent to which biomarkers from biological fluid samples reflect brain metabolism is unclear, and needs to be combined with brain metabolites to elucidate the antipyretic effect [21]. Untargeted metabolomics is a research method that systematically and comprehensively analyzes the entire metabolome, obtains and processes a large amount of metabolite data, and finds out the differential metabolites [22]. Currently, untargeted metabolomics analysis has been widely used in biomarker discovery, disease diagnosis, and mechanism research [23,24].

Although some scientific studies have previously reported on EA-inhibiting lipid peroxidation, and EA’s anti-inflammatory, antiproliferative, antiangiogenic, hyperglycemia inhibiting, and anticarcinogenic effects [25,26,27], little information is available on the use of brain metabolomics as a tool for analyzing the integral metabolites of fever with EA. Research is necessary to enable a rat brain metabolomics investigation of the pyrexia and the antipyretic anti-inflammatory effect of EA performed using Ultra-High-Performance Liquid Chromatography–Mass spectrometry (UPLC-MS). The aim of the present study was to evaluate the level of inflammatory cytokines and brain metabolite in fever, and to investigate the possible protective effect of EA on yeast-induced fever in rats.

In this study, based on untargeted metabonomics, the antipyretic effects of Asp and EA were evaluated by yeast-induced fever and pyrogenic cytokines in a rat model. Secondly, we collected brain tissue samples from rats with yeast-induced fever and performed untargeted metabolomics research on a UPLC-MS platform to explore biomarkers in rat brain. Thirdly, we used multivariate statistics and open-access databases, and filtered and identified feature metabolites. Finally, we combined the results of pathway analysis and enzyme-linked immunosorbent assay (ELISA) to reveal the thermal mechanism. Subsequently, we identified 11 metabolites associated with EA treatment, which may be involved in the regulation of brain microenvironmental states, as well as the regulation of anti-inflammatory and antipyretic pathways in the brain. These results provide more scientific evidence for the antipyretic mechanism of EA.

## 2. Results

### 2.1. Antipyretic Effects of Ellagic Acid

The rectal temperatures of rats in each group before and after drug administration were recorded to monitor the body temperature changes. Rectal temperatures and its change value were recorded and measured at 0, 3, 5, 7, 9, and 19 h after drug administration. As shown in Figure 1, the rectal temperatures and its change value of Model group (MG) were significantly higher than Normal group (NG) (*p* < 0.001), indicating that the yeast-induced pyretic model was successful after drug administration; the rectal temperatures and its change value of Aspirin group (APG) and Ellagic acid group (EAG) decreased for about 19 h. Rectal temperature and its change value were significantly decreased in APG and EAG (*p* < 0.05), indicating that EA could reduce the rectal temperature in yeast-induced fever rats.

### 2.2. ELISA Results of Inflammatory Factor IL-1β and PGE2 in Serum

Calibration curves displayed good linearity in the concentration range of 0–2000 pg/mL of IL-1β, and the concentration range of 0–1000 pg/mL of PGE2. The typical calibration curve equations and their correlation coefficients were calculated to be as follows: IL-1β, y = 86.24 + 560.02x + 221.08x^2^ (r = 0.9990); PGE2, y = (2541.27 − 2065.86x)/(1 + 3.70x + 232.51x^2^) (r = 0.9984). In the regression equation, x refers to the concentration of the analyte in serum (pg/mL), and y refers to OD of the analyte. In serum, the concentration of IL-1β in MG increased significantly compared to NG (*p* < 0.01) (Figure 2A). In the meantime, the level of IL-1β in APG and EAG decreased significantly compared to MG (*p* < 0.05). PGE2 production increased obviously because of the yeast injection. After drug administration, the levels of PGE2, APG, and EAG returned to a normal level and showed statistical significance compared to MG (Figure 2B), indicating that Asp and EA could reduce the level of proinflammatory factors in the serum of yeast-induced fever rats and has antipyretic and anti-inflammatory effects.

### 2.3. Brain Metabolomics Profile and Multivariate Data Analysis

#### 2.3.1. Principal Component Analysis

In this study, a multivariate analysis method of Principal Component Analysis (PCA) was performed to visualize the similarities and differences among four groups. The PCA scores scatter plot and loading plot are shown in Figure 3. As we can see from Figure 3A, the separation among four groups was not so good, which suggests that the perturbed metabolites in rat brain might be regulated to a normal level by applying EA. The loading plot is also called a correlation plot, the highly correlated variables are grouped together, and the inversely correlated variables are distributed at both ends of the line passing through the origins. The coordinates of each variable correspond to the correlation and directivity with PC1 and PC2, respectively; R^2^X(1) and R^2^X(2) are the explanatory rates corresponding to principal components 1 and 2. As we can see from Figure 3B, R^2^X(1) = 0.136, R^2^X(2) = 0.123, which suggests that these two principal components are not contributing very much.

#### 2.3.2. Orthogonal Partial Least Squares-Discriminant Analysis

Compared with PCA, PLS-DA can maximize the differentiation between groups and facilitate the search for differential metabolites. Orthogonal partial least squares-discriminant analysis (OPLS-DA) combines orthogonal signal correction (OSC) and the PLS-DA method, and can decompose the X matrix information into two types of information related to Y and irrelevancies. Among them, the variable information related to Y is the prediction principal component, and the variable information unrelated to Y is the orthogonal principal component. Differential variables are screened by removing irrelevant differences. The OPLS-DA model was used to analyze metabolome data and draw score charts for each group to further show the differences between each group [28]. The prediction parameters of the evaluation model are R^2^X, R^2^Y and Q^2^, where R^2^X and R^2^Y represent the explanatory rate of the model to X and Y matrix, respectively, and Q^2^ represents the prediction ability of the model. Q^2^ > 0.5 can be considered as an effective model.

The OPLS-DA scores scatter plot is shown in Figure 4. As we can see from Figure 4, a good separation was demonstrated between MG and NG, APG, and EAG, which indicates that yeast-induced fever significantly altered the levels of endogenous metabolites in rat brain. High predictability (Q^2^) of the OPLS-DA models was observed for comparisons between MG and NG (Q^2^ = 0.636, R^2^X = 0.382, R^2^Y = 0.995, *p* = 0.04, Figure 5A), as well as between MG and APG (Q^2^ = 0.789, R^2^X= 0.302, R^2^Y = 0.979, *p* = 0.05, Figure 5B), and between MG and EAG (Q^2^ = 0.532, R^2^X = 0.215, R^2^Y = 0.988, *p* = 0.005, Figure 5C). In this model, 200 random permutation and combination experiments were carried out on the data, and the model was the best when *p* < 0.05.

#### 2.3.3. Identification of Potential Biomarkers and the Variation Trends among Four Groups

Based on the results of OPLS-DA, multivariate analysis of the variable importance in projection (VIP) of the OPLS-DA model was performed to preliminarily screen out metabolites of different species or tissues; at the same time, *p*-value or fold change in univariate analysis can be combined to further screen out differential metabolites. To reveal the antipyretic mechanism of EA, two screening criteria for significant differential metabolites were established: a fold change of ≥2 or of ≤0.5 and *p*-value < 0.05; and the variable variable importance in the projection (VIP) in the OPLS-DA model of ≥1.

After completing this step of data processing, 26 metabolites, which satisfied both the criteria were selected as potential biomarkers to characterize the fever model. Detailed changing trends of these 26 metabolites among NG, MG, APG, and EAG were recorded. A total of 23 of 26 metabolites were elevated after yeast injection. Additionally, EA administration significantly regulated 12 of 26 metabolites in rat brain. It is worth mentioning that 11 of these 12 metabolites adjusted to a normal level, indicating that the 11 metabolites mentioned above may play key roles in the antipyretic effect; the results are shown in Table 1.

#### 2.3.4. Metabolic Pathway Analysis of the Potential Biomarkers

Identified biomarkers of EA antipyretic action play important roles in specific metabolic pathways. In order to determine the pathways affected by EA, we analyzed the biomarkers associated with EAG after treatment. The results showed that the potential biomarkers were responsible for the metabolic pathways in EAG were responsible for Glycerophospholipid metabolism, Sphingolipid metabolism, Phenylalanine metabolism, alpha-Linolenic acid metabolism, Glycosylphosphatidylinositol (GPI)-anchor biosynthesis, Linoleic acid metabolism, and Arachidonic acid metabolism (Figure 6), indicating that these pathways were directly related to the antipyretic mechanism of EA.

## 3. Discussion

PGE2 is considered as the main pyrogenic mediator of fever in mammals [29]. The pro-inflammatory cytokine IL-1β is thought to act as a mediator between the detection of infection stimuli and febrile stimuli by peripheral immune cells [30]. PGE2 is an important inducer of IL-1β, and its role is to establish a feedback loop to cause fever, among which IL-1β induction of PGE2 is crucial [31]. In this study, we first monitored the rectal temperature and its change value in pyretic rats, the results indicated that EA has antipyretic effect in 19 h. Moreover, the production of IL-1β and PGE2 in fever rats went significantly up after the yeast injection, and regulated after EA administration, indicating EA plays an antipyretic role by reducing central heat mediators and inflammatory factors.

Yeast-caused fever manifests as an excessive inflammatory response and metabolic disturbance in rats. EA can help decrease the body temperature by lowering inflammatory factors and regulating brain metabolism. Eleven significantly altered metabolites were selected as antipyretic biomarkers of EA in brain; in other words, EA may play an anti-fever effect by adjusting the production of 2-Hydroxyphenylacetic acid, SM (d34:1),PC (18:2 (9Z, 12Z)/P-18:1 (11Z)),PC (20:4 (8Z, 11Z, 14Z, 17Z)/P-18:0), PC (O-16:0/20:5 (5Z, 8Z, 11Z, 14Z, 17Z)), PC (16:0/16:0), PC (18:2 (9Z, 12Z)/15:0), SM (d18:1/14:0), PC (16:0/22:6 (4Z, 7Z, 10Z, 13Z, 16Z, 19Z)), PE (16:0/18:2 (9Z, 12Z)), and PC (18:1 (9Z)/18:1 (9Z)); all these metabolism are linked to glycerophospholipid metabolism and sphingolipid metabolism.

Membrane lipids contain glycerophospholipids, glycerolipids, sphingolipids, and sterol lipids [32,33,34]. Many drugs targeting lipid receptors and enzymes responsible for lipid metabolism and function have been developed and applied to a variety of diseases because of their ability to store energy, build cell membranes, molecular signals, and modify proteins [35].

Glycerophospholipids are phospholipid resynthesized by the sequential action of glycerol-3-phosphate via lysophosphatidic acid (LPA) and glycerol-3-phosphate acyltransferase (LPAAT). Phospholipases release fatty acylates from glycerophospholipids, whose diversity is produced by lysophospholipid acyltransferases (called Land’s cycle) [36,37,38,39]. Liberated fatty acyl is further metabolized by Cyclooxygenase (COXs), lipoxygenase (LOs), and cytochrome P450, resulting in production Prostaglandins, leukotrienes and epoxy fatty acids [40,41].

Sphingolipids have a sphingosine backbone with N-acyl chains and/or head groups [42,43]. Sphingosine 1-phosphate (S1P) is a lipid mediator that controls lymphocyte trafficking and is responsible for immune diseases [44,45]. S1P increases IL-1β production in human osteoblasts through the S1P1 receptor, and via the JAK and STAT3 signaling pathways [46]. The first time that S1P promotes the secretion of TNF-α and IL-1β in HLECs via S1PR1-mediated NF-κB signaling pathways, lymphangiogenesis is affected [47]. Overall, the production of Sphingolipids promotes IL-1β secretion by regulating the increase in S1P.

Based on the distribution of metabolites through metabolic pathways in rats, we produced a pathway network of EA antipyretic action (Figure 7). In yeast-induced fever in rats, eight glycerophospholipids were elevated, generating arachidonic acid with PGE2, and further generating PGE2 under the action of cyclooxygenase 2 (COX2). Excessive sphingolipids raise the expression of S1P, and promote the release of IL-1β. In our research, the level of glycerophospholipid and sphingolipid in the EAG were obviously down-regulated, consequently, the inhibition of S1P and COX2 expression inhibits the inflammatory pathways to produce thermogenic cytokines.

## 4. Materials and Methods

### 4.1. Chemicals and Reagents

Ellagic acid (EA) was purchased from Shanghai yuanye Bio-Technology Co., Ltd. (no. G26J11L119602, Shanghai, China). Yeast (Saccharomyces cerevisiae) was purchased from Sigma-Aldrich (no. BCBL8059V, St. Louis, MO, USA). ProstaglandinE2 (PGE2) was purchased from Elabscience (no. BSIB67QHK1, Wuhan, China). Interleukin-1β (IL-1β) was purchased from Elabscience (no. NW96S4YCG9, Wuhan, China).

### 4.2. Animals and Model Construction

Male Sprague-Dawley rats (150 ± 20 g) were obtained from Hunan SJA Laboratory Animal Co., Ltd. (Changsha, China). All rats were acclimated for 5 days in a controlled room with temperature (23 ± 2 °C) and humidity (55 ± 10%). The rats were kept in non-toxic, high-pressure, high-temperature, and corrosion resistant plastic cages with no more than 6 rats per cage. The rats were fed with SPF maintenance feed, and were free to eat and drink pure water.

Thirty-two healthy male rats were selected and the rectal temperature was measured twice by digital thermometer at an interval of 1 h. A total of 32 qualified rats were randomly divided into 4 groups: Normal group (NG, *n* = 8, 10 mL/kg 0.5% CMC-Na i.g.); Model group (MG, *n* = 8, 10 mL/kg 0.5% CMC-Na i.g.); Aspirin group (APG, *n* = 8, 100 mg/kg Aspirin i.g.); and Ellagic acid group (EAG, *n* = 8, 25 mg/kg Ellagic acid suspension i.g.), once a day, for consecutive 11 d. The rectal temperature of rats was measured twice, 15 min apart, and the average of the two rectal temperatures was taken as initial temperature.

The rats of MG, APG, and EAG were subcutaneously injected with 20% aqueous suspension of yeast (15 mL/kg) in their back. The rectal temperatures were measured for 0, 3, 5, 7, 9, and 19 h after the yeast injection with a digital thermometer. The rectal temperature and its change value were recorded (i.e., the difference between the rectal temperature value and the initial temperature).

The 32 animal experiments were performed under the guidelines of Animal Ethics Committee of Guangxi University of Chinese Medicine, and the experiments were approved by the Animal Ethics Committee of our university.

### 4.3. Sample Collection and Cytokines

After the last rectal temperature measurement, all rats were anesthetized by peritoneal injection of chloral hydrate. The abdominal aorta blood was immediately collected in heparinized tubes; after standing for 1 h, and centrifuged at 3000 rpm for 10 min, supernatant was obtained, separated, and stored in refrigerator at −80 °C. The rat brains were also stored at −80 °C. For measuring cytokine levels in serum, IL-1β and PGE2 ELISA kits were used by following the manufacturer’s instruction.

### 4.4. Sample Preparation and UPLC-MS Analysis

#### 4.4.1. Tissue Extraction

Samples were thawed on ice. Take 50 ± 2 mg of one sample and add cold steel balls to the mixture and homogenate at 30 Hz for 3 min. Add 1 mL 70% methanol with internal standard extract to the homogenized centrifuge tube. Whirl the mixture for 5 min, and then centrifuge it with 12,000 rpm at 4 °C for 10 min. After centrifugation, draw 400 μL of supernatant into the corresponding EP tube and store in −20 °C refrigerator overnight, then centrifuge at 12,000 r/min at 4 °C for 3 min, and take 2000 μL of supernatant in the liner of the corresponding injection bottle for on-board analysis.

#### 4.4.2. UPLC-MS Conditions

The sample extracts were analyzed using an LC-ESI-MS/MS system (UPLC, ExionLCAD, https://sciex.com.cn/, accessed on 28 June 2021; MS, QTRAP^®^ System, https://sciex.com/, accessed on 28 June 2021). The analytical conditions were as follows, UPLC: column, Waters ACQUITY UPLC HSS T3 C18 (1.8 µm, 2.1 mm × 100 mm); column temperature, 40 °C; flow rate, 0.4 mL/min; injection volume, 2 μL; solvent system, water (0.1% formic acid): acetonitrile (0.1% formic acid); gradient program, 95:5 *v*/*v* at 0 min, 10:90 *v*/*v* at 11.0 min, 10:90 *v*/*v* at 12.0 min, 95:5 *v*/*v* at 12.1 min, and 95:5 *v*/*v* at 14.0 min.

ESI source operating parameters were: source temperature 500 °C; Ion injection voltage (IS) 5500 V(positive) and −4500 V(negative); Ion source gas I (GSI), gas II (GSII), and curtain gas (CUR) were set at 55 psi, 60 psi, and 25.0 psi, respectively. Collision gas (CAD) was high. Instrument tuning and quality calibration were performed in QQQ and LIT modes in 10 and 100 μmol/L polyethylene glycol solutions, respectively. A specific set of MRM transitions for each period based on the metabolites eluted during this period was monitored.

## 5. Data Processing and Statistical Analysis

### 5.1. Data Preprocessing

Using the Metware database and the public metabolite information database, qualitative analysis was carried out on primary and secondary MS spectrum data. In the qualitative analysis of some compounds, isotopic signals, repeated signals containing K+, Na+, and NH_4_+, as well as repeated signals of fragments of other substances with larger molecular weight were removed. For analytical reference to metabolite structure, we used MassBank (http://www.massbank.jp/, accessed on 29 June 2021), KNAPSAcK (http://kanaya.naist.jp/KNApSAcK/), HMDB (http://www.hmdb.ca/, accessed on 29 June 2021) [48], MoTo DB (http://www.ab.wur.nl/moto/, accessed on 29 June 2021) and METLIN (http://metlin.scripps.edu/index.php, accessed on 29 June 2021) [49], and other public mass spectrometry databases. Metabolite quantification was completed by multiple reaction monitoring (MRM) analysis using QQQ-MS. In the MRM mode, after obtaining the spectrum analysis data of different samples, peak area integration was carried out for all peaks, and an integral correction was performed for the peaks of the same metabolite in different samples [50].

### 5.2. Principal Component Analysis

Unsupervised PCA (Principal Component Analysis) was performed by statistics function prcomp within R3.5.1 (www.r-project.org, accessed on 6 July 2021). The data were unit variance scaled before unsupervised PCA.

### 5.3. Hierarchical Cluster Analysis and Pearson Correlation Coefficients

The HCA (hierarchical cluster analysis) results of samples and metabolites were presented as heatmaps with dendrograms, while Pearson correlation coefficients (PCC) between samples were calculated by the cor function in R and presented as only heatmaps. Both HCA and PCC were carried out by R package ComplexHeatmap. For HCA, normalized signal intensities of metabolites (unit variance scaling) are visualized as a color spectrum.

### 5.4. Differential Metabolites Selected

Statistical significance of the identified metabolites was assessed using one-way ANOVA test. Therefore, *p* < 0.05 was set as the level of statistical significance. Significantly regulated metabolites between groups were determined by VIP ≥ 1, Fold Change (FC) ≥ 2 or <0.5, and *p* < 0.05. VIP values were extracted from the OPLS-DA result, which also contained score plots and permutation plots, and was generated using R package MetaboAnalystR1.0.1. The data were Fold Change (FC) and mean centered before OPLS-DA. In order to avoid overfitting, a permutation test (200 permutations) was performed.

### 5.5. KEGG Annotation and Enrichment Analysis 

Identified metabolites were annotated using the KEGG Compound database (http://www.kegg.jp/kegg/compound/, accessed on 10 July 2021), annotated metabolites were then mapped to the KEGG Pathway database (http://www.kegg.jp/kegg/pathway.html, accessed on 10 July 2021). Significantly enriched pathways were identified with a hypergeometric test’s *p*-value for a given list of metabolites [51].

## 6. Conclusions

The study established a stable and reliable UPLC/MS analytical method for rat brain metabolomics research. We first investigated the antipyretic and anti-inflammation effect of EA by measuring rectal temperature and inflammatory cytokines, respectively. Afterwards, we performed research on the brain metabolic profiling and biomarkers of yeast-induced fever in rats. Finally, we selected 26 metabolites as potential biomarkers of yeast-induced fever and 11 metabolites as biomarkers of antipyretic mechanism of EA. The involved potential target pathways are glycerophospholipid metabolism and sphingolipid metabolism. The corresponding biomarkers are PC (18:2(9Z, 12Z)/P-18:1(11Z)), PC (20:4(8Z, 11Z, 14Z, 17Z)/P-18:0), PC (O-16:0/20:5(5Z, 8Z, 11Z, 14Z, 17Z)), PC (16:0/16:0), PC (18:2(9Z, 12Z)/15:0), PC (16:0/22:6(4Z, 7Z, 10Z, 13Z, 16Z, 19Z)), PC (18:1(9Z)/18:1(9Z)), PE (16:0/18:2(9Z, 12Z)), SM (d34:1), and SM (d18:1/14:0), which can elucidate the mechanisms of antipyretic of EA to some extent. Moreover, treatment with EA suppressed the expression of IL-1β and PGE2, which are closely related to the metabolism of glycerophospholipids and sphingolipids.

These results elucidate the antipyretic mechanism of overall metabolic level of EA from the perspective of metabonomics, and provide a scientific basis for better explanation of its clinical efficacy.

## Figures and Tables

**Figure 1 molecules-27-02465-f001:**
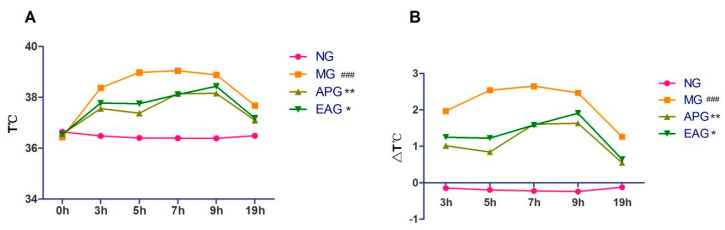
Antipyretic effects of Aspirin (100 mg/kg) and EA (25 mg/kg): (**A**). The rectal temperatures were measured at 0, 3, 5, 7, 9, and 19 h after yeast administration (*n* = 8). (**B**). Rectal temperatures change value at 3, 5, 7, 9, and 19 h after Aspirin and EA administration (*n* = 8). *p* value is for individual time point. (^###^ *p* < 0.001, vs. NG, indicates significantly different compared with the control; * *p* < 0.05, ** *p* < 0.01, vs. MG, indicate significantly different compared with the model that was incubated with 15 mL/kg yeast).

**Figure 2 molecules-27-02465-f002:**
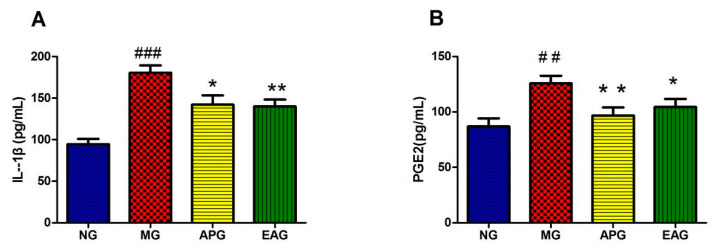
Anti-inflammation effects of Aspirin (100 mg/kg) and EA (25 mg/kg): (**A**). Effects of Aspirin and EA on the production of IL-1β (*n* = 8). (**B**). Effects of Aspirin and EA on the production of PGE2 (*n* = 8). (^##^ *p* < 0.01, ^###^ *p* < 0.001, vs. NG, indicates significantly different compared with the control; * *p* < 0.05, ** *p* < 0.01, vs. MG, indicate significantly different compared with the model that was incubated with 15 mL/kg yeast).

**Figure 3 molecules-27-02465-f003:**
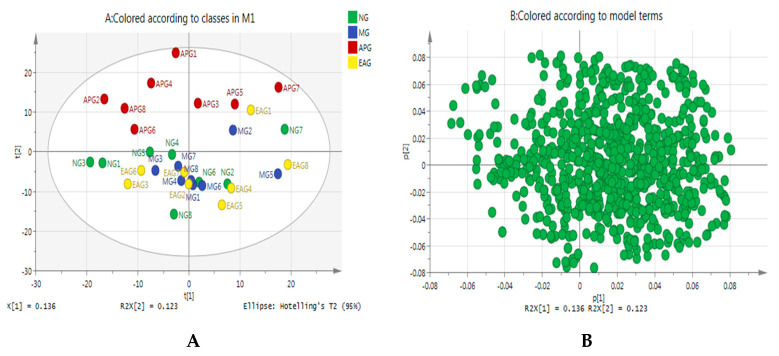
(**A**) PCA scores scatter plot of four groups; (**B**) the loading plot of four groups.

**Figure 4 molecules-27-02465-f004:**
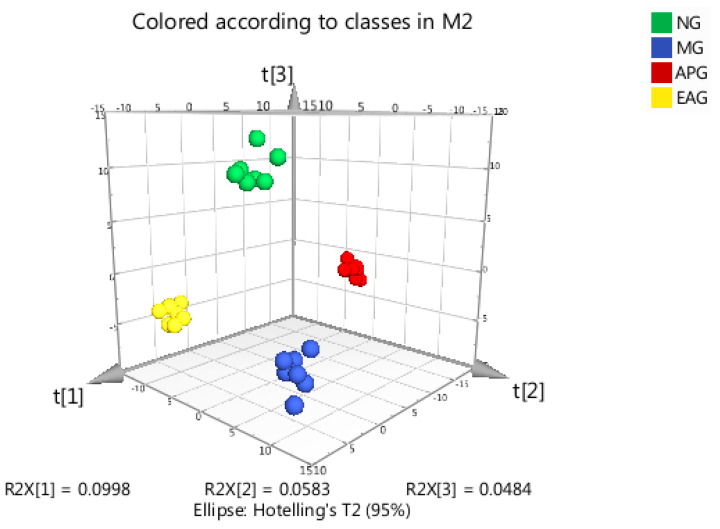
The OPLS-DA scores scatter plot of four groups.

**Figure 5 molecules-27-02465-f005:**
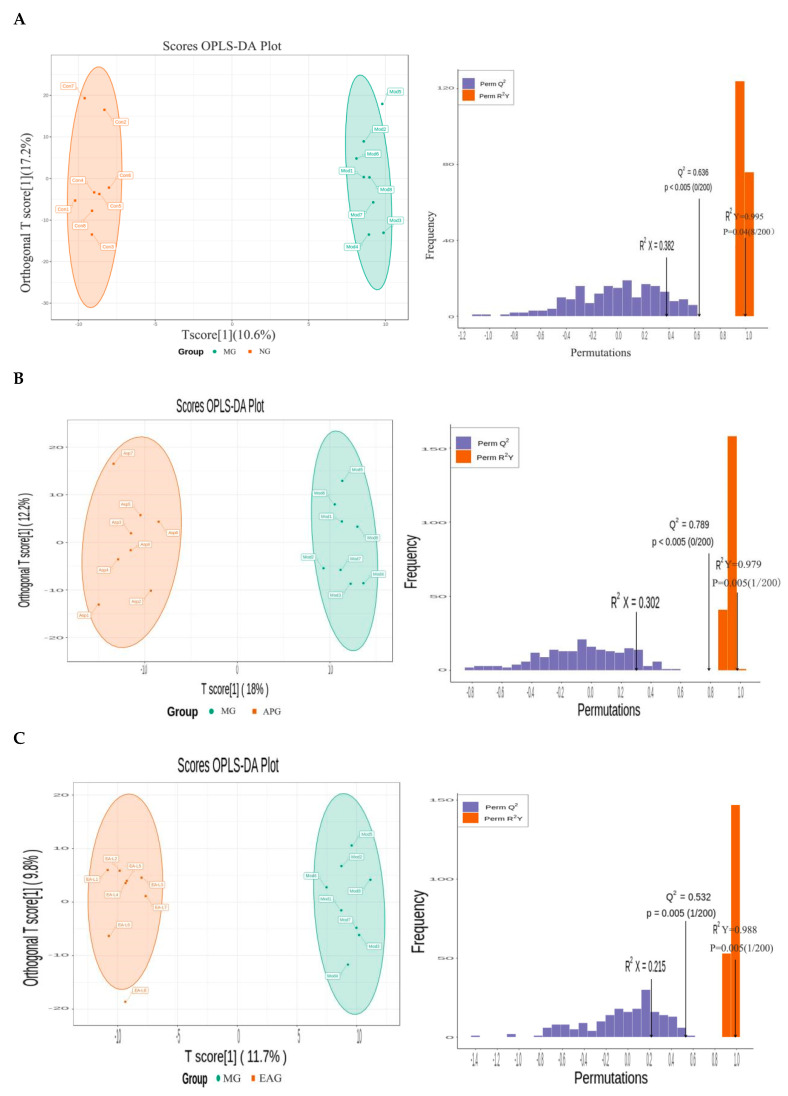
The OPLS-DA scores scatter plot: (**A**) The OPLS-DA scores of NG vs. MG. (**B**) The OPLS-DA scores of MG vs. APG. (**C**) The OPLS-DA scores of MG vs. EAG.

**Figure 6 molecules-27-02465-f006:**
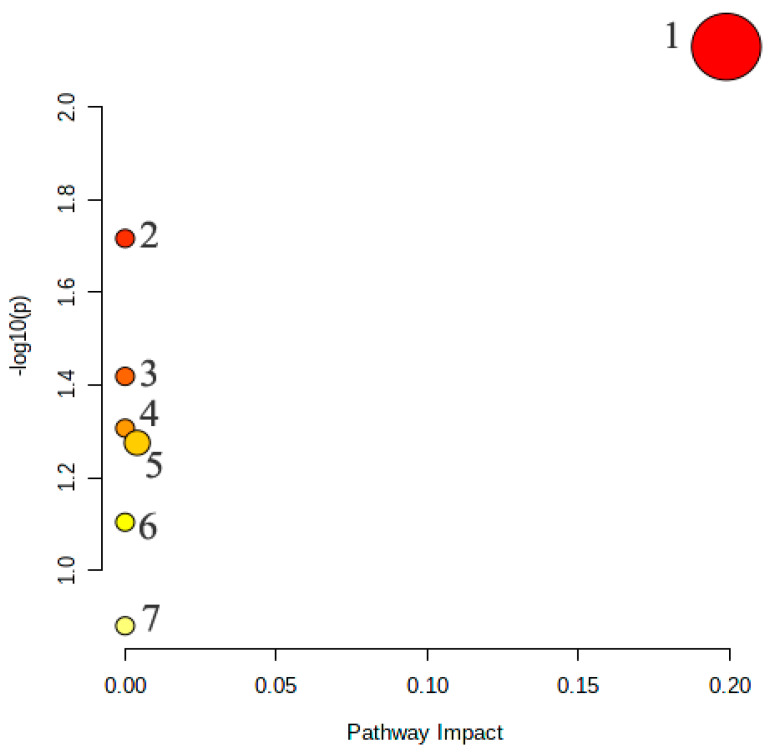
The potential Metabolic pathways in EAG: 1. Glycerophospholipid metabolism. 2. Sphingolipid metabolism. 3. Phenylalanine metabolism. 4. alpha-Linolenic acid metabolism. 5. Glycosylphosphatidylinositol (GPI)-anchor biosynthesis. 6. Linoleic acid metabolism. 7. Arachidonic acid metabolism.

**Figure 7 molecules-27-02465-f007:**
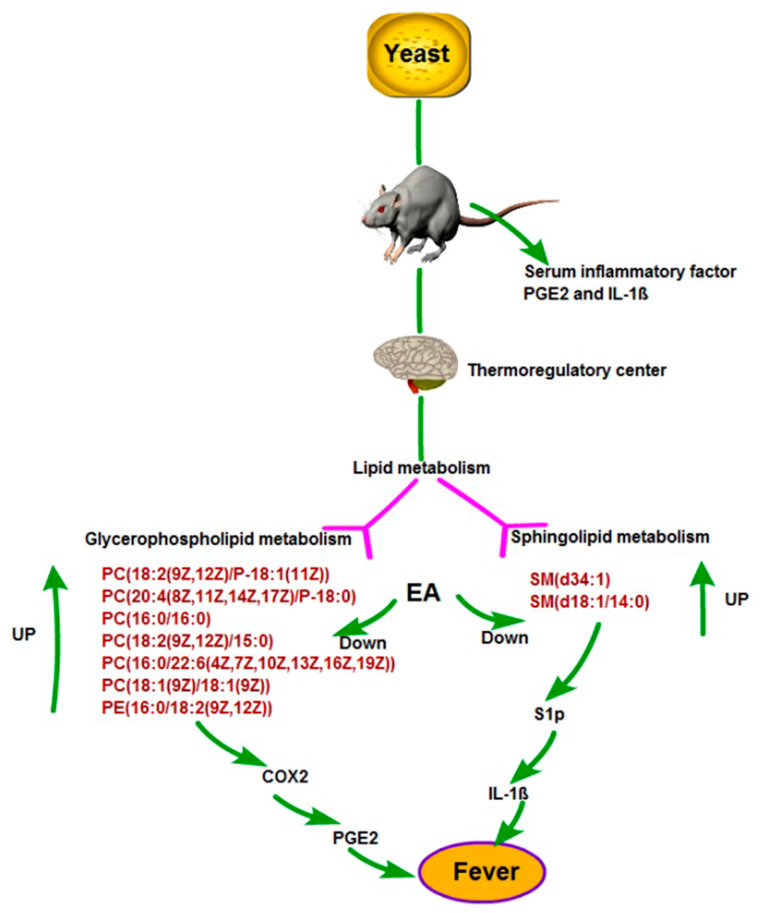
The pathways closely associated with the antipyretic effect of EA.

**Table 1 molecules-27-02465-t001:** Potential biomarkers of pyrexia.

No.	Index	Formula	Compounds	FC	NG vs. MG		MG vs. APG		MG vs. EAG	
				NG vs. MG	*p* Value	Type	*p* Value	Type	*p* Value	Type
1	MADN0334	C_8_H_8_O_3_	2-Hydroxyphenylacetic acid	2.1562	0.0399 ^#^	↓	0.2466	-	0.0148 *	↑
2	MADN0343	C_9_H_13_N_2_O_9_P	3′-UMP	0.465	0.0347 ^#^	↑	0.0151 *	↓	0.1955	-
3	MADN0408	C_10_H_15_N_2_O_8_P	Thymidine-5′-phosphate (dTMP)	0.3211	0.0128 ^#^	↑	0.0015 **	↓	0.1721	↓
4	MADN0469	C_41_H_74_NO_8_P	PE (16:0/20:4)	0.3419	0.0258 ^#^	↑	0.0865	-	0.0905	↓
5	MADN0495	C_39_H_79_N_2_O_6_P	SM (d34:1)	0.3601	0.0223 ^#^	↑	0.0322 *	↓	0.0327 *	↓
6	MADP0192	C_8_H_17_NO_2_	Methacholine	2.4518	0.0196 ^#^	↓	0.1032	-	0.0175 *	↓
7	MADP0433	C_39_H_79_N_2_O_6_P	SM (d18:1/16:0)	0.3526	0.0166 ^#^	↑	0.0500 *	↓	0.0503	↓
8	MADP0435	C_44_H_82_NO_7_P	PC (18:2(9Z, 12Z)/P-18:1 (11Z))	0.3662	0.0458 ^#^	↑	0.0577	-	0.0464 *	↓
9	MADP0436	C_46_H_84_NO_7_P	PC (20:4(8Z, 11Z, 14Z, 17Z)/P-18:0)	0.4763	0.0495 ^#^	↑	0.0437 *	↓	0.0357 *	↓
10	MADP0438	C_44_H_80_NO_7_P	PC (O-16:0/20:5 (5Z, 8Z, 11Z, 14Z, 17Z))	0.4072	0.0492 ^#^	↑	0.0488 *	↓	0.0388 *	↓
11	MADP0440	C_38_H_76_NO_8_P	PC (15:0/15:0)	0.3294	0.0411 ^#^	↑	0.0936	-	0.0505	↓
12	MADP0442	C_40_H_80_NO_8_P	PC (16:0/16:0)	0.3509	0.0317 ^#^	↑	0.0730	-	0.0430 *	↓
13	MADP0444	C_45_H_90_N_2_O_6_P^+^	SM (d16:1/24:1(15Z))	0.3685	0.0331 ^#^	↑	0.0368 *	↓	0.0641	↓
14	MADP0445	C_44_H_80_NO_8_P	PC (18:1 (9Z)/18:3 (6Z, 9Z, 12Z))	0.2118	0.0202 ^#^	↑	0.0112 *	↓	0.1082	↓
15	MADP0450	C_41_H_78_NO_8_P	PC (18:2 (9Z, 12Z)/15:0)	0.4158	0.0342 ^#^	↑	0.0747	-	0.0141 *	↓
16	MADP0452	C_45_H_91_N_2_O_6_P	SM (d18:1/22:0)	0.3602	0.0364 ^#^	↑	0.0830	-	0.0713	↓
17	MADP0455	C_37_H_75_N_2_O_6_P	SM (d18:1/14:0)	0.4935	0.0149 ^#^	↑	0.0207 *	↓	0.0278 *	↓
18	MADP0459	C_41_H_81_N_2_O_6_P	SM (d18:1/18:1(9Z))	0.2787	0.0245 ^#^	↑	0.0380 *	↓	0.0565	↓
19	MADP0468	C_46_H_80_NO_8_P	PC (16:0/22:6 (4Z, 7Z, 10Z, 13Z, 16Z, 19Z))	0.4856	0.0483 ^#^	↑	0.0754	-	0.0328 *	↓
20	MADP0469	C_42_H_80_NO_8_P	PC (16:0/18:2 (11Z, 13Z))	0.3724	0.0461 ^#^	↑	0.0569	-	0.0600	↓
21	MADP0470	C_41_H_83_N_2_O_6_P	SM (d18:1/18:0)	0.3249	0.0298 ^#^	↑	0.0708	-	0.0559	↓
22	MADP0472	C_39_H_74_NO_8_P	PE (16:0/18:2(9Z, 12Z))	0.3882	0.0353 ^#^	↑	0.0994	-	0.0355 *	↓
23	MADP0473	C_48_H_84_NO_8_P	PC (18:0/22:6 (4Z, 7Z, 10Z, 13Z, 16Z, 19Z))	0.3796	0.0466 ^#^	↑	0.1106	-	0.0625	↓
24	MADP0475	C_44_H_84_NO_8_P	PC (18:1 (9Z)/18:1 (9Z))	0.3025	0.0252 ^#^	↑	0.0126 *	↓	0.0334 *	↓
25	MADP0478	C_44_H_86_NO_8_P	PC (18:0/18:1 (9Z))	0.3128	0.0439 ^#^	↑	0.0899	-	0.1194	↓
26	MEDP1928	C_10_H_16_N_2_O_2_	Cyclo (Pro-Val)	2.7492	0.0000 ^##^	↓	0.3157	-	0.6655	-

^#^ *p* < 0.05, ^##^
*p* < 0.01, vs. NG; * *p* < 0.05, ** *p* < 0.01 vs. MG. ↑: up regulated; ↓: down regulated.

## Data Availability

The data presented in this study are available on request from the corresponding author. The data are not publicly available due to the first author is still a PhD student and has not graduated yet, relevant research is continuing.

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
