# Peer review of "The Potential Antipyretic Mechanism of Ellagic Acid with Brain Metabolomics Using Rats with Yeast-Induced Fever"

_molecules, 2022, doi:10.3390/molecules27082465_

Round 1

Reviewer 1 Report

In abstract, .26 metabolites; should be: twenty six

In Introduction, the novelty should be addressed.

The specific objective of this study is recommended to be stated in the last Introduction part.

In methods, the number of ethical clearance should be stated

The validity of ELISA in terms of validation criteria (sensitivity, precision, etc.) for analysis of r IL-1β and PGE2 in Serum.

In PCA results, the authors should also discussed the loading plot to evaluate the contribution of each variables for such differentiation.

The validation model of OPLS-DA should be highlighted. 

Author Response

Thank you very much for the suggestions of reviewers,i will make some explanations for my modify:

1.In abstract,26 has been changed from twenty-six;

2.In Introduction, the novelty has been added;

3.The specific objective of this study has been added;

4.In methods, the number of ethical clearance has been stated

5.The validity of ELISA in terms of validation criteria for analysis of r IL-1β and PGE2 in Serum has been added;

6.In PCA results, the loading plot has been added and the contribution of each variables for such differentiation is described;

7.The validation model of OPLS-DA has been introduced. 

Reviewer 2 Report

The manuscript describes the use of untargeted metabolomic approach to understand the effects of ellagic acid on the global metabolite changes in the brain during yeast-induced fever. Although the conclusion is not surprising, the conclusion of the manuscript is supported by the results.

Suggestions:

The authors need to state that they used untargeted metabolomic approach as no lipid standards are used.

Authors need to state the version of metaboanalyst being used

Is the raw datasets available for public access?

Error bar is needed for Figure 1. Please also state the amount of aspirin and ellagic acid being used in the figure legend. Also state the n number

Is figure 1A and 1B the same results, just another way of presentation? If so, only keep one.

What does the p value mean for figure 1? It is for individual time point or over 19 hrs

For figure 2, more experimental details is needed in the figure legend.

Figure 4. The font size in the figure need to be increased as it is currently unreadable

Please state the parameters being used in metaboanalyst 

The UPLC-MS data are run on positive or negative ion mode?

Author Response

Thank you very much for the suggestions of reviewers,i will make some explanations for my modify:

1.The authors add a description of the use of an untargeted metabonomics approachï¼›

2.Authors state the version of metaboanalyst being used

3.The raw datasets available for public accessï¼›

4.The legend in Figure 1 has been modifiedï¼›

5.The results in Figure 1A and Figure 1B are different,Figure 1A is rectal temperatures and Figure 1B is rectal temperatures change value,it is indicated in the legend;

6.p value is for individual time point,it is indicated in the legend;

7.The text size of Figure 4 has been increased;

8.Parameters used in metabolic analysis have been added in 5.Data Processing and Statistical Analysis 

9.The UPLC-MS data are run on positive and negative ion mode,it has been stated in 4.4.2. UPLC-MS Conditions.

Round 2

Reviewer 1 Report

The authors have responded my comment. therefore, the paper can be accepted as its current form.

Reviewer 2 Report

Authors addressed my questions adequately 

This manuscript is a resubmission of an earlier submission. The following is a list of the peer review reports and author responses from that submission.